# Position: Multi-Agent Systems Should Prioritize Concurrency Control

Xin Yang [* 1]  Letian Li [* 2]  Zimo Ji [3]  Terry Jingchen Zhang [† 4]  Wenyuan Jiang [† 4]

## Abstract

LLM-based multi-agent systems (MAS) promise scalable collaboration, yet adding agents often *reduces* reliability. This position paper argues that many MAS failures are fundamentally **concurrency control problems**: agents concurrently read and write shared state, and long LLM inference windows amplify the risk of stale reads, lost updates, and inconsistent outcomes. Failure modes commonly attributed to "coordination" or "communication" breakdowns can be mapped directly onto classical concurrency anomalies. We contend that MAS frameworks should address these failures through explicit concurrency control mechanisms: conflict detection, isolation guarantees, and structured access to shared resources. Concurrency control should be a first-class design concern, not an afterthought.

## 1. Introduction

Large language models (LLMs) have demonstrated increasingly powerful capabilities in language understanding, reasoning, and generation (Brown et al., 2020; OpenAI, 2023; Touvron et al., 2023; Yang et al., 2026), expanding their application domains from text generation to complex decision-making. The integration of tool calling and environmental interaction has further transformed LLMs into autonomous agents capable of executing real-world tasks through iterative reasoning and action (Yao et al., 2023; Schick et al., 2023; Mialon et al., 2023; Cheng et al., 2025). This paradigm shift, exemplified by the ReAct framework (Yao et al., 2023), enables models to move beyond pure dialogue toward practical task completion through observation, reasoning, and action loops. More recently, the pursuit of

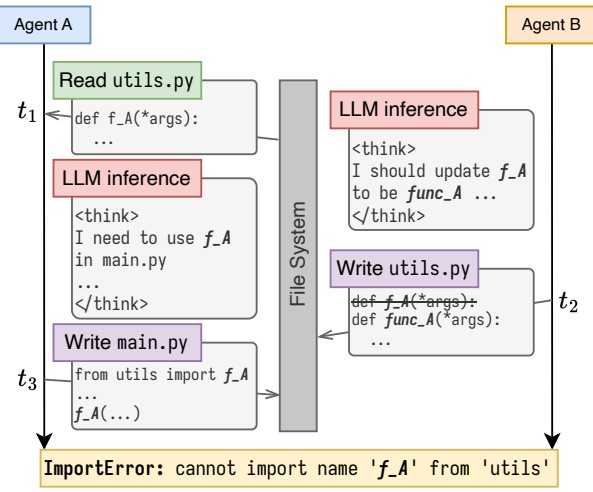

*Figure 1.* **Stale read hazard in multi-agent coding.** Agent A reads `utils.py` and enters a long inference phase while implementing `main.py`. Concurrently, Agent B refactors `utils.py`, renaming f_A into func_A. Both agents act correctly in isolation, yet the interleaving yields a broken import, a classic concurrency anomaly amplified by long LLM inference windows.

solving increasingly complex tasks has driven the development of multi-agent systems (MAS), where multiple agents operate *concurrently*, communicating and collaborating to achieve goals at greater scale and with higher success rates than single-agent approaches (Park et al., 2023; Li et al., 2023; Hong et al., 2024; Qian et al., 2024; Wu et al., 2023; Guo et al., 2024; Anonymous, 2026). However, empirical evidence reveals a sobering reality: scaling the number of agents does not reliably improve performance (Cemri et al., 2025; Kim et al., 2025). Studies show that multi-agent systems exhibit failure rates between 41% and 86.7% across popular benchmarks, with coordination failures and inter-agent misalignment accounting for a substantial fraction of these breakdowns (Cemri et al., 2025; Li et al., 2024a). Recent systems and benchmarks provide concrete evidence that concurrency mechanisms affect MAS outcomes (Geng & Neubig, 2026; Chen et al., 2024a; Zhang et al., 2026; Wang et al., 2024b; Chang & Geng, 2025).

**In this position paper, we argue that many communication and coordination failures in MAS are fundamentally**

---
[*]Equal contribution  [1]School of Mathematical Sciences, Zhejiang University, Hangzhou, China [2]Shenzhen International Graduate School, Tsinghua University, Shenzhen, China [3]Hong Kong University of Science and Technology, Clear Water Bay, Hong Kong [4]ETH Zurich, Zurich, Switzerland. Correspondence to: Wenyuan Jiang <wenyjiang@ethz.ch>.

*Proceedings of the 43rd International Conference on Machine Learning*, Seoul, South Korea. PMLR 306, 2026. Copyright 2026 by the author(s).

**concurrency control problems, which should be considered as a prioritized bottleneck for building efficient, scalable multi-agent systems.**

Our claim targets MAS in which agents read or modify shared mutable state during long inference windows, including shared repositories, blackboard memories, message buffers, and embodied world states. Systems with disjoint inputs face lower concurrency risk and may be bottlenecked by reasoning, planning, or communication quality. This scope complements MAS surveys that organize workflows, infrastructure, and collaboration patterns (Chen et al., 2024b; Li et al., 2024b; Tran et al., 2025; Guo et al., 2024); our contribution is the concurrency-control lens that connects independently developed coordination mechanisms.

*Table 1.* Recent evidence that concurrency mechanisms affect MAS outcomes.

| Source | Concurrency signal | Reported effect |
|---|---|---|
| CAID | worktree isolation, merge validation | 63.3% isolated vs. 55.5% unisolated; single-agent 57.2% |
| CodeR | dependency scheduling | 22% vs. 10% resolved after removing the task graph |
| Silo-Bench | barriers, state conflicts | 67.1% of failures; RCC reaches 100% at high contention |
| MegaAgent | parallel scheduling | 800s vs. 4505s without parallel group execution |
| SagaLLM | saga transactions | correct reactive planning where baseline LLM planners fail |

*Table 2.* Concurrency-attributable failures are a substantial fraction.

| Failure Mode | Source | Rate | Concurrency Root |
|---|---|---|---|
| Premature submission | Silo-Bench | 37.2% | Missing sync barriers |
| Consensus failure | Silo-Bench | 29.9% | Concurrent conflicting states |
| Inter-agent misalignment | MAST | 36.9% | Stale reads, inconsistent state |
| Coordination overhead | Silo-Bench | RCC $\leq$ 100% | Concurrency scaling penalty |

To illustrate this perspective concretely, consider a motivating example involving two coding agents collaboratively developing software on a shared file system. Agent $A$ reads an existing utility module `utils.py` and subsequently implements a new feature in `main.py` that imports and invokes a function `f_A` from the utility module. Concurrently, Agent $B$ is refactoring `utils.py`, renaming `f_A` to `func_A`. Unfortunately, between the moment $A$ reads the utility module and the moment $A$ completes its implementation, $B$ writes the refactored version. The result: both agents complete their individual tasks correctly from their own perspectives, yet the system enters an inconsistent state where `main.py` contains broken imports. This failure superficially appears to be a coordination or communication problem. However, examining it through the lens of concurrent systems reveals a classic concurrency hazard: a *stale read* leading to an inconsistent state.

What makes such hazards particularly prevalent in MAS is a fundamental *temporal asymmetry*: LLM inference time (e.g., the duration of the "thinking" phase) is typically orders of magnitude longer than the execution time of tool actions, dramatically expanding the window during which interleaved operations can produce conflicts. While an agent reasons for seconds or even minutes, other agents may modify the shared environment multiple times, invalidating the assumptions underlying the first agent's decisions.

The remainder of this paper is organized as follows. We first demonstrate that common MAS failure modes can be systematically understood as **concurrency hazards** (§2), revealing that diverse "coordination problems" share a common structure rooted in concurrent access to shared state. We then present **recommendations** for concurrency control in MAS design (§3, Table 3), addressing objectives, system-level mechanisms, and trade-offs among correctness, efficiency, and scalability. Finally, we discuss open challenges and issue a **call to action** (§5) for cross-disciplinary collaboration between ML and systems-oriented communities.

## 2. Concurrency Hazards in Disguise

Many failures in multi-agent systems, often framed as coordination or communication issues, can be understood as classic concurrency hazards. We present representative failure cases and map them to established concurrency control concepts, showing that these challenges mirror problems long studied in systems and database research (Bernstein & Goodman, 1981; Bernstein et al., 1987; Weikum & Vossen, 2002). This perspective motivates treating concurrency control as a core design principle for MAS and grounding coordination heuristics in explicit concurrency semantics.

### 2.1. Failure Examples

We present four representative failure scenarios in multi-agent systems that are often attributed to poor coordination or communication, but are more precisely explained as classic concurrency hazards over shared state.

**Stale Read.** In a collaborative coding MAS (Qian et al.,

*Table 3.* Design space for MAS concurrency control. Trade-offs evaluated against: task success (**S**), compatibility (**C**), efficiency (**E**), inference cost (**I**). Arrows: ↑ improves, ↓ degrades.

| Layer | Decision | Options (Trade-offs) |
|---|---|---|
| *System Design* | Isolation level | Weak/Read Committed (E↑, S↓: more parallelism, risks anomalies) ↔ Strong/Serializable (S↑, E↓) |
| | Control strategy | Pessimistic/locking (S↑, E↓: blocks during long inference) vs. Optimistic/validation (E↑, I↓: wastes compute on abort) |
| | Versioning | Single-version (simple) vs. MVCC (E↑: readers never block writers; C↓: added complexity) |
| | Transaction granularity | Fine-grained/single action (E↑, I↓: shorter conflicts, higher overhead) vs. Coarse/subtask (I↑, S↓: expensive rollbacks) |
| | Transaction boundaries | Explicit `BEGIN`/`COMMIT` (C↑, I↓: flexible, requires model understanding) vs. Implicit/system-inferred (I↑, C↓) |
| | Lock/resource granularity | Coarse/files (I↑, E↓) vs. Fine/functions (E↑, I↓: more parallelism, more metadata) |
| *Infrastructure* | Backend system | Custom (C↑: tailored semantics) vs. Existing DB/Git/FS (S↑, C↑: mature guarantees, may not fit agent semantics) |
| | Version control integration | Branch-per-subtask (S↑: isolation; E↓: merge overhead) vs. Validation-at-merge (E↑, S↓: deferred conflict detection) |
| | Inference optimization | Standard vs. Optimized batching/speculation/quantization (E↑, S↑: shorter transactions reduce conflict window) |
| | Checkpointing | None (simple) vs. KV-cache checkpointing (I↑: efficient rollback without full recomputation; C↓: engine support required) |
| *Model* | Concurrency training | None vs. SFT/RL on conflict scenarios (S↑: better anticipation/resolution; I↓: requires data and compute) |
| | Prompt intervention | Generic vs. Concurrency-aware prompts (I↑: low cost; S±: limited, brittle guarantees) |
| | Task decomposition | Overlapping resources (E↑, S↓) vs. Disjoint partitioning (S↑, C↓: requires upfront design effort) |
| | Failure feedback | Opaque "retry" (I↑: simple) vs. Semantic conflict details (S↑, I↓: enables adaptation, requires model capability) |

2024; Hong et al., 2024), Agent $A$ reads a configuration file `config.yaml` and implements a client module based on the observed endpoints. Concurrently, Agent $B$ updates the configuration to reflect new infrastructure. When $A$ later writes its code, it relies on outdated assumptions, resulting in an inconsistent system state despite both agents acting correctly in isolation. This is a standard stale-read anomaly.

**Lost Update.** Two agents independently modify different parts of a shared file `utils.py`. Agent $A$ optimizes one function, while Agent $B$ fixes a bug in another. Because both read the same initial version and write back independently, the later write overwrites the earlier one, silently discarding one agent's contribution. This lost update occurs without either agent detecting a conflict.

**Stale Correction.** In message-based MAS (Li et al., 2023; Wu et al., 2023), Agent $A$ broadcasts a plan, Agent $B$ sends a correction, but Agent $C$ begins executing the original plan before receiving the update. Although the correction is sent, it is not applied atomically with respect to all agents, leading to incorrect execution driven by outdated information.

**Action-Message Desynchronization.** In embodied environments such as Minecraft-based benchmarks (Fan et al., 2022; Wang et al., 2024a; Dong et al., 2024), agents coordinate via both messages and world-altering actions. An agent may act on a message describing an intended state that has already been invalidated by another agent's concurrent action. As a result, agent beliefs and the true environment state diverge, even when communication is logically consistent.

**Implicit Assumptions in Existing Systems.** These hazards are present in current MAS architectures. Systems such as MAGIS (Tao et al., 2024) improve reliability through structured orchestration and post-hoc Git-based merging, but implicitly assume benign interleavings during concurrent reasoning. Conflicts are detected only after agents have already performed expensive inference on incompatible assumptions. This reflects a broader pattern across MAS frameworks that rely on orchestration while leaving concurrency semantics underspecified. Recent systems show the same pattern from another angle. CAID uses isolated worktrees and merge-time validation, CodeR uses a task graph to order dependent work, SagaLLM uses saga-style compensation, and MegaAgent emphasizes parallel scheduling (Geng & Neubig, 2026; Chen et al., 2024a; Chang & Geng, 2025; Wang et al., 2024b). These mechanisms correspond to optimistic isolation, dependency scheduling, transactional recovery, and concurrent scheduling, even when they are introduced without explicit concurrency-control terminology.

## 2.2. A Unified Framework

The failure examples above, though arising in different application contexts, share a common structure. We now develop

a unified framework that reveals these failures as violations of fundamental concurrency properties, drawing on classical concurrency control theory (Bernstein et al., 1987; Weikum & Vossen, 2002; Adya, 1999).

**Formalizing Multi-Agent Systems.** We model a multi-agent system as a collection of $n$ agents with respective internal states $\{\mathcal{A}^1, \mathcal{A}^2, \ldots, \mathcal{A}^n\}$, all operating within a *shared* environment $\mathcal{E}$. Each agent $j$ executes a sequence of actions $a_1^j, a_2^j, \ldots$ according to its own local ordering, but these sequences interleave concurrently across agents, i.e., actions may overlap temporally, and conflicting operations on shared resources can occur.

We classify actions based on whether they modify the shared environment. An action is *side-effect-free* (a *read*) if it leaves $\mathcal{E}$ unchanged; examples include reading a file, observing game state, or retrieving messages. An action is *side-effecting* (a *write*) if it modifies $\mathcal{E}$; examples include writing a file, placing a game object, or broadcasting a message.

We further distinguish between *agent-local state* and *environment state* based on *mutability by other agents*: environment state can be modified by any agent's writes, while agent-local state (e.g., context window, reasoning trace) can only be directly modified by that agent itself. In coding MAS, the shared file system constitutes environment state. In message-passing MAS, each agent's mailbox, which is writable by other agents via message sends, is environment state. In game environments, both world state and communication channels are environment state.

A key observation is that LLM-based reasoning is side-effect-free with respect to $\mathcal{E}$, but requires substantial time to complete. This *temporal asymmetry*, i.e., inference spans seconds to minutes while tool execution completes in milliseconds, dramatically expands the window during which interleaved operations can produce conflicts.

**Consistency and Isolation.** We identify two properties whose violation underlies MAS failures, drawing on the ACID properties from database theory (Härder & Reuter, 1983):

*Consistency* requires that concurrent operations do not leave the system in a conflicting or contradictory state. In MAS, this means the combined effects of all agents' actions should be mutually compatible: two agents should not simultaneously assume exclusive use of the same resource, and actions taken based on observations should remain valid when executed. Consistency is a *global* property concerning system-wide coherence.

*Isolation* requires that each agent's execution proceeds "as if" it were the only agent in the system. Formally, the outcome should be equivalent to some serial execution of all

agents' actions (Weikum & Vossen, 2002). Isolation is an *agent-centric* property: no agent should observe another agent's incomplete operation sequence. The classical notion of *serializability*, that concurrent execution should be equivalent to some sequential ordering, captures the strongest form of isolation (Papadimitriou, 1979).

These properties are complementary: consistency ensures global coherence, while isolation ensures local predictability. Violations of either lead to the anomalies described above.

**Mapping Failures to the Framework.** In *coding MAS*, the *write-after-read* failure violates both consistency (Agent $A$'s implementation conflicts with Agent $B$'s configuration changes) and isolation (Agent $A$'s read was invalidated by Agent $B$'s concurrent write). The *lost update* failure similarly violates both: consistency is violated because only one agent's modifications persist, and isolation is violated because one agent's write was silently overwritten.

In *message-based MAS*, the *stale correction* failure maps directly to the write-after-read pattern: Agent $C$ reads the original plan, Agent $B$ writes the correction, but Agent $C$'s actions proceed based on the stale read. In *cooperative game MAS*, the *action-message desynchronization* failure violates consistency because the coordination message and world state become inconsistent, and isolation because Agent $B$'s operation is disrupted by Agent $C$'s uncoordinated action.

**Implications for MAS Design.** The unified framework identifies failures documented across diverse MAS applications as systematic consequences of uncontrolled concurrency. The database and distributed systems communities have developed extensive techniques to address precisely these problems: locking protocols (Eswaran et al., 1976), optimistic concurrency control (Kung & Robinson, 1981), multi-version concurrency control (Reed, 1978), and various isolation levels (Berenson et al., 1995; Adya, 1999). For MAS research, the central task is adapting classical concurrency control mechanisms to the unique characteristics of LLM-based agents: long inference latencies, natural-language interfaces, and imperfect adherence to specifications. We address this task in the following section.

## 3. Concurrency Control for MAS

Having established that many MAS failures stem from concurrency hazards, we now address the central question: how should MAS designers approach concurrency control? This section articulates design objectives and examines the mechanisms available to system architects, organized across three complementary layers (Table 3): system design, infrastructure and model capabilities. Together, these layers provide a comprehensive framework for understanding where concurrency control responsibilities reside and how they may be systematically addressed.

## 3.1. Design Objectives

Effective concurrency control for MAS should balance multiple, often competing objectives. We identify four primary dimensions that collectively define the quality of a concurrency control solution.

**Task Success Rate.** The fundamental objective is ensuring that concurrent agent execution produces correct results. What constitutes "success" depends on the application: in coding tasks, it may mean passing all tests without introducing bugs; in collaborative writing, it may mean producing coherent output without contradictory edits. A concurrency control mechanism should maximize task success rate across diverse MAS workloads. Existing MAS benchmarks (Liu et al., 2024b; Cemri et al., 2025; Jimenez et al., 2024; Zhou et al., 2024) provide starting points for measuring success, though they largely ignore concurrency-specific failure modes.

**Compatibility.** Concurrency control mechanisms should accommodate diverse MAS architectures: centralized orchestrators (Wu et al., 2023), decentralized peer-to-peer designs (Li et al., 2023), hybrid topologies (Hong et al., 2024), and hierarchical structures (Qian et al., 2024). A mechanism tightly coupled to a specific communication pattern or agent architecture limits reusability. Ideally, concurrency control primitives should be orthogonal to the MAS coordination strategy, enabling modular integration without redesigning the entire system.

**Execution Efficiency.** Concurrency control inherently restricts parallelism to ensure correctness, creating tension with performance goals. Key efficiency metrics include: (i) *end-to-end task completion time*: the wall-clock duration for the MAS to finish a task; (ii) *effective parallelism*: the degree to which agents operate concurrently without blocking; and (iii) *abort/retry rate*: the frequency of transaction rollbacks due to detected conflicts. An ideal mechanism minimizes completion time while maintaining correctness guarantees.

**Inference Cost.** MAS concurrency control incurs costs beyond those of traditional distributed systems: direct overhead from tool calls for lock acquisition or conflict detection; model comprehension cost, as LLMs may struggle with concurrency primitives and require extended prompts or fine-tuning; and context window consumption from concurrency-related instructions and state tracking. Critically, mechanisms requiring agents to explicitly manage locks or reason about conflicts demand far greater model sophistication than transparent system-level enforcement.

## 3.2. System Design for MAS

We now examine the theoretical foundations and concrete design choices for MAS concurrency control, spanning isolation levels, control strategies, primitives, and integration with broader MAS mechanisms.

### 3.2.1. THEORETICAL FOUNDATIONS

**Isolation Levels.** Database systems formalize correctness through *isolation levels*, which specify the degree to which transactions are protected from interference (Berenson et al., 1995; Adya, 1999). The classical hierarchy includes, from weakest to strongest: Read Uncommitted, Read Committed (RC), Repeatable Read (RR), Snapshot Isolation (SI), Serializable (SER), and Strict Serializable (SSER) (Bernstein et al., 1987; Fekete et al., 2005; Papadimitriou, 1979). Each level prevents progressively more anomalies (e.g., dirty reads, non-repeatable reads, phantom reads, and write skew) but typically incurs higher overhead (Cahill et al., 2009).

For MAS, the appropriate isolation level depends critically on the application domain. A multi-agent coding system modifying a shared codebase may require serializability to prevent silent corruption from conflicting edits. In contrast, a multi-agent debate system where agents operate on largely disjoint logical resources (their own arguments) may tolerate weaker guarantees like Read Committed, trading strict isolation for higher throughput. The general idea is that stronger isolation provides better guarantees to agents but may reduce performance and increase implementation complexity, which is a tradeoff that should be calibrated per application.

**Pessimistic vs. Optimistic Control.** Concurrency control strategies fall into two broad categories (Bernstein & Goodman, 1981; Kung & Robinson, 1981). *Pessimistic* approaches, such as two-phase locking (2PL), acquire locks before accessing resources, blocking other agents until locks are released (Eswaran et al., 1976; Gray et al., 1976). This prevents conflicts proactively but may cause deadlocks (circular waits) and reduces parallelism. *Optimistic* approaches allow agents to proceed without locks, validating at commit time that no conflicts occurred; conflicting transactions abort and retry. Optimistic methods suit low-contention workloads but may thrash under high contention.

A crucial distinction from traditional systems is the *temporal asymmetry* in MAS: LLM inference typically spans seconds to minutes, while simple tool calls complete in milliseconds. This asymmetry fundamentally alters the calculus. Under pessimistic control, a lock held during LLM reasoning blocks other agents for extended periods, severely degrading parallelism. Under optimistic control, aborts waste substantial compute: an agent may reason for minutes only to have its transaction invalidated. Neither approach dominates; the choice depends on contention levels, task structure, and the relative costs of blocking versus retry. Recent coding-agent evidence illustrates this trade-off: CAID reports that isolated branches with merge-time validation

outperform a shared-workspace multi-agent baseline, suggesting that optimistic isolation is effective when conflicts can be validated cheaply at integration time (Geng & Neubig, 2026).

**Multiversion Concurrency Control.** Multiversion concurrency control (MVCC) offers a middle ground by maintaining multiple versions of data (Reed, 1978; Bernstein et al., 1987). Readers access consistent snapshots without blocking writers, enabling the principle that "readers never block writers and writers never block readers." MVCC naturally supports Snapshot Isolation (Fekete et al., 2005) and forms the basis for concurrency control in many modern databases. For MAS, MVCC-style approaches could allow agents to observe consistent environment states while others make modifications, with conflict detection deferred to commit time. The challenge lies in defining "versions" for arbitrary agent environments beyond traditional databases.

### 3.2.2. Primitive Design

**Transaction Granularity.** A core design decision is the unit of atomicity. Transactions may range from individual actions (e.g., a single tool call) to entire agent subtasks (e.g., "implement feature X"). Fine-grained transactions shorten conflict windows but incur higher coordination overhead, while coarse-grained transactions reduce overhead at the cost of higher conflict probability and more expensive rollbacks.

Another question is whether transaction boundaries are *explicit*, via primitives such as BEGIN/COMMIT, or *implicit*, inferred by the system from agent behavior or task structure. Explicit control gives agents flexibility but assumes that LLMs can reliably reason about transactional semantics. Implicit control lowers cognitive burden but risks poorly chosen boundaries that either fragment logical work or bundle too much state into a single transaction.

**Resource Abstraction and Lock Types.** Lock-based designs should define what constitutes a lockable resource: entire files, individual functions, database rows, or higher-level semantic objects (e.g., "a calendar slot"). A *fine-grained resource* is the smallest unit to which access control applies. In a codebase, file-level locking treats utils.py as one resource, while function-level locking treats parse_config and validate_input as distinct resources. Function-level locking allows two agents to modify different functions concurrently, whereas file-level locking serializes them unnecessarily. This mirrors row-level and table-level locking in databases (Gray et al., 1976). Finer-grained resources enable more parallelism but require more complex tracking and metadata.

Lock modes further regulate access (Gray et al., 1976). *Shared (read) locks* permit concurrent observation, while *ex-clusive (write) locks* serialize modifications. This distinction aligns well with MAS workloads, where reads are frequent and typically safe to overlap, whereas writes should be carefully coordinated. More expressive mechanisms, such as intention or predicate locks, may be useful for hierarchical or semantic resources.

**Conflict Detection and Rollback.** Optimistic control requires detecting conflicts by tracking *read sets* and *write sets*, then validating that no incompatible updates occurred concurrently (Kung & Robinson, 1981; Adya, 1999). Software transactional memory (STM) offers a reference model for such tracking (Shavit & Touitou, 1995; Harris et al., 2005), though MAS environments are more heterogeneous than the shared-memory settings STM targets. This heterogeneity is an analytical distinction: MAS resources extend beyond memory locations to files, prompts, tool outputs, mailboxes, external APIs, and natural-language commitments. Beyond conflict detection within the MAS runtime itself, recent black-box and history-based database isolation checkers suggest a complementary direction: treating MAS executions as observable histories and using an external monitor to check whether interactions satisfy different consistency or isolation levels (Tan et al., 2020; Liu et al., 2024a; Cai et al., 2025).

When conflicts arise, aborted transactions should be rolled back. In MAS, rollback is challenging because agent state includes external writes, internal context, and intermediate reasoning. Techniques such as KV-cache checkpointing (Kwon et al., 2023) may allow partial rewinding without full re-inference. However, actions with irreversible side effects (e.g., sending emails or invoking external APIs) fundamentally limit rollback and should be treated with special care.

### 3.2.3. MAS Integration

**Synchronization Granularity.** MAS frameworks should decide on the *minimal synchronization unit*: the finest granularity at which agents coordinate. One approach uses *round-based synchronization*: agents act in discrete rounds, with a barrier ensuring all agents complete round $k$ before any begins round $k + 1$. This simplifies reasoning about concurrent state but limits parallelism and may be inefficient when agents' tasks have heterogeneous durations. The alternative is *fully asynchronous* execution where agents proceed independently, relying solely on concurrency control primitives (locks, validation) for coordination. This maximizes parallelism but requires more sophisticated conflict handling.

The synchronization granularity should align with transaction boundaries. If transactions span multiple rounds in a round-based system, mid-transaction preemption creates complex partial states. Conversely, if transactions complete within single rounds, the round barrier naturally provides

commit points.

**Inference Interruption.** A distinctive aspect of MAS is whether ongoing inference can be interrupted. An agent mid-reasoning may be holding logical locks or accumulating reads that will be validated at commit. If another agent's commit invalidates these reads, should the first agent be *preempted* (immediately aborted) or allowed to complete then fail validation? Preemption avoids wasted compute but requires mechanisms to safely interrupt LLM inference and restore state. Allowing completion then validation is simpler but wastes resources when conflicts are predictable.

Related questions include whether agents can *voluntarily* yield execution (cooperative scheduling) and how interrupted agents resume from the beginning of the transaction, from a checkpoint, or with partial context preserved via KV cache.

**Semantic Feedback on Failures.** When a transaction fails due to conflict detection, lock timeout, or validation failure, the agent receives an error signal. A key design choice is whether to provide *semantic explanations* of failures. A rich explanation might state: "your write to `utils.py` conflicted with Agent B's concurrent modification of the `parse_config` function." An opaque signal simply indicates "transaction aborted, please retry."

Semantic feedback enables agents to reason about conflicts and adapt strategies—perhaps avoiding the contested resource, coordinating with the conflicting agent, or decomposing the task differently. However, this requires models capable of interpreting and acting on such information, a capability that may require specific training or careful prompting. Opaque signals are simpler but provide no guidance for conflict resolution.

**Integration with Existing Pipelines.** A practical question is how to augment existing MAS workflows, which typically assume benign interleavings, with explicit concurrency semantics. Consider systems like MAGIS (Tao et al., 2024) that use structured role specialization and Git-based collaboration. Several "drop-in" strategies could add concurrency guarantees without redesigning the entire pipeline:

• *Branch-per-subtask*: Each agent subtask operates on an isolated Git branch, with merges gated by automated conflict detection and semantic validation (e.g., test execution, type checking).
• *Transactional file edits*: Wrap multi-file modifications in lightweight transactions that abort if concurrent changes to the same files are detected before commit.
• *Validation-at-merge*: Defer validation to merge time and enrich merge checks with semantic constraints beyond textual diff resolution.

These strategies trade off eager conflict prevention against deferred conflict detection. Branch-per-subtask maximizes isolation but complicates cross-branch dependencies; validation-at-merge preserves existing workflows but may waste computation on branches that ultimately conflict.

### 3.3. Infrastructure Support

MAS designers can leverage existing infrastructure that already provides well-tested concurrency primitives.

**File Systems and Databases.** File systems offer atomic operations and advisory locking, while relational databases provide full ACID transactions with configurable isolation levels (Härder & Reuter, 1983; Weikum & Vossen, 2002). Vector databases increasingly expose transactional semantics as well (Pinecone Systems, 2023; Weaviate, 2023). Structuring agent interactions around such systems allows MAS to inherit mature concurrency guarantees and performance optimizations, offloading much of the correctness burden.

**Version Control Systems.** Version control systems such as Git support concurrent work via branching and explicit merge-based conflict resolution. Recent work explores Git as coordination infrastructure for MAS. EvoGit (Huang et al., 2025) uses Git's DAG structure to enable asynchronous multi-agent development, isolating agents on branches and synchronizing via merges while avoiding global serialization. This design favors scalability and fault isolation, but defers semantic conflict resolution to merge time, where textual diffs may not capture program-level invariants.

**Context Management Infrastructure.** Git-like abstractions have also been applied to agent context management. GCC (Wu, 2025) provides branching and merging for agent memory, enabling efficient context reuse and performance gains. However, it targets primarily single-agent settings and does not define concurrency semantics for multi-agent interleavings over shared context.

Across these systems, a common pattern emerges: infrastructure supplies powerful *mechanisms* (transactions, branches, merges), while higher-level frameworks must still define *policies*: when to isolate, what conflicts are acceptable, and how to enforce semantic invariants. Bridging this gap between infrastructure capabilities and concurrency semantics is central to the agenda we advocate.

**Inference Engine Efficiency.** The temporal asymmetry that exacerbates MAS concurrency issues (i.e., long inference latencies relative to tool execution) can be mitigated by faster inference. Advances including continuous batching (Yu et al., 2022b), paged attention (Kwon et al., 2023), speculative decoding (Leviathan et al., 2023), and quantization (Frantar et al., 2022) reduce per-token latency and thus shrink the window during which an agent holds logi-

cal locks or accumulates stale reads. From a concurrency control perspective, faster inference is equivalent to shorter transactions, directly reducing conflict probability.

**Efficient Checkpointing for Rollback.** Optimistic concurrency control requires efficient rollback. In LLM inference, this translates to restoring the KV cache to a prior state. Modern inference engines with paged KV cache management (Kwon et al., 2023) can potentially support *forking* (copying cache state for speculative execution) and *rollback* (discarding cache entries added after a checkpoint). Explicit support for these operations would enable efficient transaction abort without full recomputation. Cache sharing across agents with common prefixes (Zheng et al., 2024) further reduces the cost of retry by amortizing prompt processing.

### 3.4. Model Capabilities

Complementing system-level approaches, the underlying LLMs can be improved to exhibit concurrency-aware behavior through evaluation, training, and inference-time interventions.

#### 3.4.1. BENCHMARKING CONCURRENCY AWARENESS

A prerequisite for improving model capabilities is measuring them. We advocate for benchmarks that evaluate: (i) an agent's ability to *anticipate* conflicts given knowledge of other agents' goals and current activities; (ii) an agent's ability to *detect* conflicts from environmental feedback (e.g., failed writes, stale reads); and (iii) an agent's ability to *resolve* conflicts through negotiation, backing off, or requesting arbitration. Such benchmarks could extend existing MAS evaluation suites (Liu et al., 2024b; Cemri et al., 2025) with concurrency-specific metrics: conflict rate, resolution success rate, wasted computation from aborts, and throughput under contention.

A particularly valuable benchmark type would measure agent behavior *without* explicit concurrency control, revealing how naturally (or poorly) current models handle concurrent settings and what failure modes emerge most frequently.

#### 3.4.2. TRAINING FOR CONCURRENCY AWARENESS

Given appropriate benchmarks and environments, LLMs can be trained to exhibit concurrency-aware behavior. Supervised fine-tuning (SFT) on traces of successful conflict anticipation, detection, and resolution can instill basic patterns. Reinforcement learning (RL) in multi-agent environments, building on the substantial MARL literature (Lowe et al., 2017; Foerster et al., 2018; Yu et al., 2022a; Zhang et al., 2019), can train agents to coordinate under contention, learning when to yield, retry, or escalate.

The reward signal should reflect both task success and co-ordination efficiency, penalizing unnecessary conflicts and wasted retries. Centralized training with decentralized execution can mitigate non-stationarity when agents share rewards and training infrastructure. The harder case is heterogeneous deployment, where agents from different providers or training regimes coordinate without prior joint training, a setting closely related to Ad Hoc Teamwork (Stone et al., 2010; Barrett & Stone, 2015). In such deployments, as one agent's policy or system prompt changes, the effective environment for other agents shifts, complicating convergence and reliable coordination (Hernandez-Leal et al., 2017). Techniques from cooperative MARL, including centralized training with decentralized execution and opponent modeling, may help address this challenge.

#### 3.4.3. INFERENCE-TIME INTERVENTIONS

Without modifying models, some concurrency-aware behavior can be induced through prompt design and task structuring.

**Prompt Engineering.** Agents can be explicitly informed that they operate in a concurrent environment with shared resources and guided toward conservative behaviors, such as checking for concurrent edits, acquiring resources in a consistent order, or retrying after failed writes. Such prompts rely on the base model's instruction-following ability and whatever concurrency knowledge it has implicitly acquired during pretraining, and thus offer limited and brittle guarantees.

**Task Decomposition and Orchestration.** Contention can often be reduced by decomposing tasks so that agents operate on largely disjoint resources, with hierarchical approaches dividing responsibilities between strategic and tactical agents to reduce coordination complexity. While perfect partitioning is rarely possible, careful decomposition can substantially lower conflict frequency. These approaches trade upfront design effort for reduced runtime conflicts, but scale poorly to novel or highly interdependent workloads where explicit concurrency control becomes necessary.

## 4. Alternative Views

**Foundation model capabilities alone suffice.** One view holds that sufficiently capable models can resolve conflicts without explicit concurrency control. Conversation-centric frameworks such as AutoGen exemplify the appeal of relying on model capability and dialogue structure for coordination (Wu et al., 2023). However, as agent count grows, conflict probability approaches certainty regardless of model quality. Natural-language negotiation wastes inference budget, coordination embedded in model weights is neither debuggable nor auditable, and robust concurrency aware-

ness would require training data that does not exist. Model behavior also remains non-deterministic in practice (Atil et al., 2024; Kahng et al., 2024).

**Traditional systems techniques suffice.** Another view argues that existing database transactions and locks can be directly applied to MAS. However, LLM agents are semantic-driven, non-deterministic (Atil et al., 2024), and incur inference costs orders of magnitude larger than system-level operations. These mismatches make direct transplantation ineffective and instead call for system–model co-design.

**Eventual consistency and convergence suffice.** A third view suggests that eventual consistency (e.g., via CRDTs) is sufficient (Shapiro et al., 2011; Kleppmann & Beresford, 2017). This conflates convergence with semantic correctness. While replicas may eventually agree, application invariants can be violated when agents reason over long inference windows based on invalidated state. CRDTs suit independent operations with well-defined merge semantics, but MAS often rely on cross-resource invariants that should remain stable during reasoning. The key question is what guarantees agents can rely on while they think and act.

**Existing community tools already cover the issue.** Existing tool (SE branching and CI, MARL coordination, and distributed consistency protocol) are valuable starting points, but LLM-MAS uniquely combines shared-state hazards with long, costly, language-mediated inference. The open problem is exposing these as agent-facing primitives with clear isolation, validation, and rollback semantics.

# 5. Call to Action

Many coordination failures in multi-agent systems are, at their core, concurrency control problems. The anomalies well-known from parallel programs and distributed databases resurface when agents share mutable state, amplified by long LLM inference windows and non-determinism. Closing this gap requires cross-disciplinary collaboration. ML researchers should treat concurrency as a first-class systems concern rather than emergent behavior to be prompted around, while researchers in PL and DB should adapt classical techniques to LLM-based agents through careful redesign. Neither perspective alone is sufficient.

## 5.1. For ML Researchers

**Recognize concurrency as a distinct failure mode.** When MAS exhibit timing-dependent errors or coordination breakdowns despite locally correct actions, the root cause is often concurrent access to shared state under long inference windows. Treating these as generic coordination problems obscures the underlying issue. Explicit recognition is the first step toward systematic solutions.

**Design benchmarks with contention in mind.** Current evaluations (Liu et al., 2024b; Jimenez et al., 2024; Zhou et al., 2024) report aggregate success rates but cannot distinguish capability failures from concurrency failures. A system achieving 70% success may fail 20% of the time due to stale reads alone, yet this remains invisible in current metrics. New benchmarks should vary contention levels systematically and measure conflict frequency, resolution success, wasted computation from aborts, and effective parallelism under contention.

**Train for concurrency awareness.** Pretraining corpora contain extensive coverage of sequential programming but minimal exposure to concurrent coordination patterns such as lock denial, abort recovery, or conflict anticipation under partial observability. Whether models can acquire robust concurrency-aware behavior via prompting alone or require explicit training with multi-agent RL (Lowe et al., 2017; Foerster et al., 2018) remains an open and consequential question.

## 5.2. For Researchers in Systems-Oriented Fields

**Adapt, not transplant.** Databases and distributed systems offer mature foundations: serializability theory (Papadimitriou, 1979), isolation hierarchies (Berenson et al., 1995; Adya, 1999), and protocols such as 2PL (Eswaran et al., 1976), OCC (Kung & Robinson, 1981), MVCC (Reed, 1978), and SSI (Cahill et al., 2009). However, LLM agents interact through natural language and are non-deterministic, making direct transplantation ineffective. The core challenge is determining which abstractions transfer, which require redesign, and what new primitives are needed.

**Design for latency asymmetry.** Database transactions complete in microseconds; LLM inference takes seconds to minutes. Pessimistic locking blocks unacceptably at these timescales; optimistic control wastes expensive computation on aborts. New protocols should explicitly account for this asymmetry, potentially via interruptible inference, KV-cache rollback, or contention-aware scheduling. The success criterion is preserving task success while improving effective parallelism and reducing wasted tokens under controlled contention.

**Build agent-friendly concurrency infrastructure.** MAS currently lack the primitives that make concurrency manageable in traditional systems: versioning, conflict detection, and rollback semantics exposed through interfaces that agents can reliably use. Equally important are observability and debugging tools that make concurrency failures explicit and actionable. Without such tooling, diagnosing whether a failure originates from a race condition, a coordination misalignment, or a model-level error remains prohibitively difficult in practice. Building this infrastructure is a prerequisite for principled progress.

## Acknowledgements

We thank Dr. Si Liu at ETH Zürich for constructive discussions on concurrency control topics.

## Limitations

While this position paper establishes a compelling conceptual mapping between MAS failures and classical concurrency anomalies, our framework primarily targets systems with explicitly shared mutable state, and may not generalize to MAS architectures that rely on implicit coordination or emergent communication. Additionally, the trade-offs among correctness, efficiency, and inference cost are discussed qualitatively; more rigorous empirical validation across diverse real-world workloads is left for future work.

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

## A. LLM Usage

We used large language models (LLMs) solely for writing assistance, including grammar checking, language polishing, and phrasing refinement. All technical content, experimental design, and conclusions are entirely the work of the authors.

