# OpenReview forum: "Position: Multi-Agent Systems Should Prioritize Concurrency Control"
_ICML.cc/2026/Position_Paper_Track — ICML 2026 Position Paper Track regular_

### Official Review · Reviewer_cwD4 · 2026-03-06

**Significance:** 2
**Argument Clarity:** 2
**Rating:** 3
**Confidence:** 3

**Questions:**

1. Could you explain to me what the fine-grained resource is?

**Alternative Views Section:**

Yes

**Compliance With Llm Reviewing Policy A Conservative:**

Affirmed.

**Discussion Potential:**

2

**Final Justification:**

This paper is an interesting position paper with a good writing. However, from the perspective of MARL/multi-agent systems literature I do not think this position paper is with strong relation to multi-agent systems. However, I do not deny that it establishes a clear connection between the distributed database systems and the current LLM systems. To this end, I keep my original score, but I do not resist to the acceptance of this paper.

**Paper Summary:**

This paper primarily shows a position that the failures of LLM-based multi-agent systems in communication and coordination are mainly caused by the concurrency control problems, and people should take this as a priority in design. The main contributions of this paper are: (1) it thoroughly introduces the common failure examples of LLM-based multi-agent systems to motivate the relation to concurrency control problems; (2) it establishes an overall picture about how concurrency control can used to improve LLM-based multi-agent systems, from multiple dimensions; (3) finally, it answers to some potential alternative views, and calls people from various fields to participate in the action of developing concurrency control.

**Position:**

Yes

**Position In Title:**

Yes

**Related Work:**

2

**Strengths And Weaknesses:**

Strengths:
1. This position paper is in general well-written with a clear position and a good structure to expand this position through different dimensions, which is generally easy to follow.
2. The viewpoints about concurrency control or database theories are very well stated with clear and strong evidence to support them.
3. Related work about the traditional view of concurrency control for file operations and database management is well cited.

Neutral:
1. From my own perspective, it is unclear if the position of this paper can inspire some discussion. The reason is file operation itself is part of LLM engineering. Since I am not an expert in LLMs, I am not sure if people have not considered this point. Although the authors have claimed the difference between the traditional database operation strategies and the ones they propose, I still feel like the one proposed in this position paper is a natural case of distributed database systems corresponding to LLM-based multi-agent systems.
2. About the fitness to the ICML community, I think it is controversial. The reason is that the position is weakly related to learning itself. Instead, it is more tended to give a position about how we can improve a system with some software engineering techniques. Although the authors have mentioned some contents about multi-agent reinforcement learning, the proportion is too low across the whole paper. However, due to the current nature of LLMs, I am not sure if this is suitable for ICML.

Weaknesses:
1. Some claims about LLMs are with no clear reference or evidence to support:
	1. L389-391: One view holds that sufficiently capable models can resolve conflict without explicit concurrency control.
	2. L407-409: A third view suggests that eventual consistency (e.g., via CRDTs) is sufficient.
	3. L311-315: Structuring agent interactions around such systems allows MAS to inherit mature concurrency guarantees and performance optimizations, offloading much of the correctness burden.
	4. L259-261: though MAS environments are more heterogeneous than the shared-memory settings STM targets.
2. In L222-243, for the two paragraphs: Transaction Granularity and Resource Abstraction and Lock Types, I cannot understand what the fine-grained resource is. Also, the examples about LLMs should be with reference to support them, otherwise, I do not know if these are your own imagination or facts.
3. In section 3.4.2, the authors have covered the potential issue and solution about multi-agent reinforcement learning. They claimed non-stationarity problem. However, if I understand the position clearly, all of claims are mainly built upon shared resources to improve synchronisation. In this situation, centralised training naturally fit and is always the first priority, so there is actually no issue of non-stationarity.
4. In my own viewpoint, the position is almost irrelevant to multi-agent systems. Instead, it is more proper to be categorised into introducing concurrency control or database operation techniques to improve LLMs. I do not think this will weaken the value of this position paper. For example, I cannot understand the purpose of "Formalizing Multi-Agent Systems". Without this, I think the follow-up paragraphs can still well reflect the position.

**Support:**

2

---

> ### Author Rebuttal · Authors · 2026-03-28
>
> We thank the reviewer for the detailed reading and specific feedback. We address each point below. (All new citations will be added in the revised paper.)
>
> **R4-W1: Some claims about LLMs have no clear reference.**
> We will add specific references:
> - L389 ("models suffice"): Wu et al. (2024, AutoGen) exemplify relying on model capability for coordination without explicit concurrency control.
> - L407 ("CRDTs"): Shapiro et al. (2011) formalize CRDTs; Kleppmann & Beresford (2017) apply them to collaborative editing.
> - L311 ("inherit guarantees"): follows from leveraging existing DB infrastructure with ACID properties (Haerder & Reuter, 1983).
> - L259 ("more heterogeneous"): contrast between shared-memory STM (Harris et al., 2005) and MAS environments with NL interfaces -- we will explicitly mark this as our analytical observation rather than an empirical claim.
>
> **R4-W2: Cannot understand what the fine-grained resource is.**
> A fine-grained resource is the smallest unit to which access control applies. Concretely: if Agents A and B both modify `utils.py` but touch different functions (`parse_config` vs. `validate_input`), function-level locking allows both to proceed concurrently, whereas file-level locking serializes them unnecessarily. This mirrors row-level vs. table-level locking in databases (Gray, 1976). Finer granularity enables more parallelism but requires more metadata. We will add this example with a formal definition in the revision.
>
> **R4-W3: Centralized training naturally fits, so no non-stationarity.**
> When agents are trained centrally with shared rewards (CTDE), non-stationarity is mitigated -- we agree with the reviewer. Our discussion in Sec 3.4.2 considers a different setting: deployment where heterogeneous agents from different providers (e.g., GPT-4 + Claude + open-source models) must coordinate in a shared environment without joint training. In this increasingly common scenario, independent training makes non-stationarity relevant. We will explicitly scope this claim to the decentralized/heterogeneous deployment setting to avoid confusion.
>
> **R4-W4: Position is almost irrelevant to multi-agent systems; more proper as CC for LLMs.**
> We respectfully disagree. Concurrency hazards arise specifically from *multi-agent structure*: a single agent with exclusive environment access does not face stale reads, lost updates, or consensus failures (Sec 1, formalized in Sec 2.2). The formalization shows why these hazards emerge from concurrent multi-agent access to shared state, not from any individual agent's limitations.
>
> More importantly, while individual coordination techniques exist across SE, DB, and ML communities, no prior work connects them as instances of the same concurrency control problem in the MAS context. Recent systems independently reinvent CC patterns: CAID uses git-worktree isolation (= optimistic CC), SagaLLM applies saga transactions (= DB recovery), CodeR builds task dependency graphs (= scheduling), MegaAgent implements parallel scheduling -- yet none cite concurrency control literature. Our paper provides the unifying theoretical framework that explains *why* these systems converge on similar solutions and how classical CC theory can guide future MAS design more systematically.
>
> The temporal asymmetry (seconds-to-minutes inference vs. millisecond tool calls) distinguishes LLM-MAS from traditional distributed systems and requires adapted rather than transplanted solutions -- this is not "natural" distributed DB but a fundamentally new regime.
>
> **R4-N1: Unclear if this can inspire discussion; natural case of distributed DB.**
> See R4-W4 above. The key distinction is that distributed DB assumes deterministic, fast operations. LLM agents are non-deterministic, slow (orders of magnitude), and communicate via natural language. These differences invalidate core assumptions of classical protocols (e.g., lock duration, abort cost, retry semantics) and demand new solutions at the intersection of ML and systems. The fact that multiple ML systems are already reinventing DB techniques ad hoc -- without the theoretical vocabulary to diagnose or compare their approaches -- demonstrates exactly the discussion our position aims to inspire.
>
> **R4-N2: Fitness to the ICML community.**
> Our paper addresses learning questions at the intersection of ML and systems: can models acquire concurrency-aware behavior through training (Sec 3.4.2), what MARL formulations support coordination under resource contention (Sec 3.4.2), how should benchmarks measure concurrency awareness as a model capability (Sec 3.4.1), and how does inference-time behavior interact with system-level control (Sec 3.4.3). The Position Track is well suited for cross-disciplinary perspectives that reframe existing problems -- here, showing that fragmented MAS coordination solutions share a common theoretical foundation in concurrency control.

---

> > ### Author Rebuttal · Reviewer_cwD4 · 2026-04-01
> >
> > **"We respectfully disagree. Concurrency hazards arise specifically from multi-agent structure: a single agent with exclusive environment access does not face stale reads, lost updates, or consensus failures (Sec 1, formalized in Sec 2.2). The formalization shows why these hazards emerge from concurrent multi-agent access to shared state, not from any individual agent's limitations."**
> >
> > I agree that the word "multi-agent structure" is more appropriate than multi-agent systems to summarize your work. This is actually what I felt like that your work mainly built on the structure with multiple identities.
> >
> > **When agents are trained centrally with shared rewards (CTDE), non-stationarity is mitigated -- we agree with the reviewer. Our discussion in Sec 3.4.2 considers a different setting: deployment where heterogeneous agents from different providers (e.g., GPT-4 + Claude + open-source models) must coordinate in a shared environment without joint training.**
> >
> > The setting you mentioned is called Ad Hoc Teamwork.
> >
> > **The key distinction is that distributed DB assumes deterministic, fast operations. LLM agents are non-deterministic, slow (orders of magnitude), and communicate via natural language. These differences invalidate core assumptions of classical protocols (e.g., lock duration, abort cost, retry semantics) and demand new solutions at the intersection of ML and systems. The fact that multiple ML systems are already reinventing DB techniques ad hoc -- without the theoretical vocabulary to diagnose or compare their approaches -- demonstrates exactly the discussion our position aims to inspire.**
> >
> > I am a MARL person, and have no knowledge about ML systems. For this reason, I cannot justify the statement you provide.

---

> > > ### Author Response · Authors · 2026-04-03
> > >
> > > We sincerely thank the reviewer for the thoughtful follow-up and for the helpful clarifications. We especially appreciate your comments on the phrasing of *"multi-agent structure"* and for pointing out the connection to *Ad Hoc Teamwork*. These insights help us better align our terminology with the MARL literature, and we will incorporate them explicitly in the revision.
> > >
> > > ---
> > >
> > > **(1) On grounding the setting in existing literature**
> > >
> > > We agree that more clearly situating our setting within existing literature will improve the paper. In particular:
> > >
> > > * The heterogeneous coordination scenario we consider is closely related to **Ad Hoc Teamwork** [1,2], where agents must collaborate without prior joint training.
> > > * The non-stationarity issue we discussed is well-established in MARL [3], especially under independently trained agents and dynamically changing teammates.
> > >
> > > We will make these connections explicit and add the corresponding references to better contextualize our claims.
> > >
> > > ---
> > >
> > > **(2) On the distinction from distributed databases**
> > >
> > > We agree that there is a conceptual connection to distributed database systems. Our main point is that LLM-based settings introduce practical differences-such as long inference time, non-determinism, and language-mediated state-that make direct application of classical concurrency control non-trivial and call for adaptation rather than straightforward reuse.
> > >
> > > ---
> > >
> > > **(3) On cross-community justification**
> > >
> > > We appreciate your perspective from the MARL side, and we agree that the systems-related motivation should be made more concrete and accessible. In the revision, we will strengthen this aspect in two ways.
> > >
> > > First, we will include clearer, concrete examples of existing LLM-based multi-agent systems that already exhibit patterns analogous to concurrency control, even if not explicitly framed as such. For instance, Git-based workflows introduce isolation via branching, while validation-at-merge resembles optimistic concurrency control. These patterns appear across recent systems, suggesting that similar solutions are being rediscovered in practice without a unifying abstraction.
> > >
> > > Second, we will clarify that our contribution is primarily conceptual rather than empirical: we aim to provide a **unifying lens** that connects these emerging system designs to well-established concurrency control theory. The goal is not to claim novelty in individual mechanisms, but to explain *why* these patterns arise and how they can be more systematically understood, compared, and extended.
> > >
> > > We hope this clearer positioning helps bridge the gap between system-oriented insights and the MARL perspective, making the argument easier to evaluate without requiring deep familiarity with ML systems work.
> > >
> > > ---
> > >
> > > **(4) Clarifying scope based on your feedback**
> > >
> > > Finally, we are encouraged that there appears to be alignment on key aspects of the framing-particularly regarding the use of *multi-agent structure* and the relevance of *Ad Hoc Teamwork* to the heterogeneous setting. We will reflect these points more clearly in the revision, which we hope addresses a substantial part of the remaining concerns.
> > >
> > > ---
> > >
> > > Thank you again for the constructive feedback-it has been very helpful in improving both the clarity and positioning of the paper.
> > >
> > > [1] Stone P, Kaminka G, Kraus S, et al. (2010). Ad hoc autonomous agent teams: Collaboration without pre-coordination. AAAI.
> > >
> > > [2] Barrett S, Stone P. (2015). Cooperating with unknown teammates in complex domains: A robot soccer case study of ad hoc teamwork. AAAI.
> > >
> > > [3] Hernandez-Leal P, Kaisers M, Baarslag T, et al. (2017). A survey of learning in multiagent environments: Dealing with non-stationarity. arXiv:1707.09183.

---

### Official Review · Reviewer_pxB7 · 2026-03-10

**Significance:** 3
**Argument Clarity:** 3
**Rating:** 4
**Confidence:** 2

**Questions:**

Please see the above question above.

**Alternative Views Section:**

Yes

**Compliance With Llm Reviewing Policy A Conservative:**

Affirmed.

**Discussion Potential:**

3

**Paper Summary:**

The paper presents the standpoint that the current concurrency problem in multi-agent systems is a key source of failures in such systems. The paper provides several pieces of evidence to support this claim. Accordingly, it proposes a new framework to control concurrency, inspired by successes in the systems design community that reduce latency in interactions between different entities. The paper also presents alternative views, including using a single LLM, relying on traditional system design, or ignoring concurrency evaluation, and argues that these views may not present the full picture. In the end, the paper offers several suggestions for the machine learning and systems communities to advance the development of multi-agent systems based on these findings.

**Position:**

Yes

**Position In Title:**

Yes

**Related Work:**

2

**Strengths And Weaknesses:**

Strength
* This paper is clearly written and easy to follow.
* The paper presents a key and emerging research question for the community.
* The reasoning behind this question is clearly argued through different levels of diagnosis.
* The paper presents a detailed framework to tackle the concurrency problem and also includes a section on alternative views.

Weakness
* The discussion on the alternative views could be more thorough given the ongoing debate around MAS. Currently, many points are mentioned but mostly at a high level, particularly for the transitional system design part. Also, the alternative views are framed mainly in terms of whether one agrees with the perspective of this paper, whereas broader alternative views could also reflect how different communities view MAS systems.
* The call-to-action section does not clearly incorporate how the framework proposed in this paper for concurrency control in MAS could inspire future work. It focuses more on raising awareness and evaluating concurrency, but the connection between the proposed designs and concrete future actions seems incomplete.
* A more explicit related work section would be very helpful in positioning this paper within the current literature.
* It is unclear how significant the quantitative effect of concurrency is; for example, in real-world applications, how much delay agents experience in their communications is not clearly discussed.

**Support:**

3

---

> ### Author Rebuttal · Authors · 2026-03-28
>
> We thank the reviewer for the positive assessment and thoughtful suggestions. We address each point below. (All new citations will be added in the revised paper.)
>
> **R3-W1: Alternative views could be more thorough.**
> We will expand to engage substantively with three community perspectives beyond the current framing: (a) **Software engineering**, where branch-based workflows and CI/CD pipelines are standard practice -- our contribution is arguing these should be first-class MAS primitives, not just deployment infrastructure; (b) **MARL**, where decentralized execution under partial observability addresses related but distinct coordination problems -- concurrency control complements rather than replaces these approaches by targeting shared-state hazards specifically; (c) **Distributed systems**, where consistency-availability tradeoffs (CAP) constrain achievable guarantees -- we will discuss how these tradeoffs manifest differently under LLM latency asymmetry, where "partition" is replaced by "stale inference window." Each perspective will be treated as a genuine alternative rather than solely as a contrast to our position.
>
> **R3-W2: Call-to-action connection to proposed framework seems incomplete.**
> We will strengthen the mapping from our design space (Table 1) to concrete future work: "Design for latency asymmetry" -> investigate MVCC with KV-cache snapshots (Sec 3.1-3.2); "Build agent-friendly infrastructure" -> expose transactional APIs in existing MAS frameworks (Sec 3.3); "Train for concurrency awareness" -> develop benchmarks measuring conflict anticipation and resolution (Sec 3.4.1). Each call-to-action will be tied to a specific row in Table 1 with explicit success criteria.
>
> **R3-W3: A more explicit related work section.**
> We currently integrate related work as each concept is introduced. We will add an explicit positioning paragraph in Sec 1 relative to MAS surveys (Chen et al., 2024; Li et al., 2024; Tran et al., 2025) and concurrent systems work, clarifying that our contribution is the concurrency-control lens connecting independently developed coordination solutions, rather than a new survey of MAS.
>
> **R3-W4: Unclear how significant the quantitative effect of concurrency is.**
> We consolidate concurrency-related measurements across recent studies:
>
> | Concurrency Effect | Source | Metric |
> |:--|:--|:--|
> | Scaling overhead | Silo-Bench (Zhang'26) | SR drops 70% (N=2->100); RCC=100% at N>=50 |
> | Barrier/consensus failures | Silo-Bench (Zhang'26) | 67.1% of failures from missing barriers + state conflicts |
> | Isolation ablation | CAID (Geng'26) | With isolation 63.3%, without 55.5% (< single-agent 57.2%) |
> | Coordination removal | CodeR (Chen'24) | 22%->10% resolved (-55%) without task graph |
> | Sequential penalty | MegaAgent (Wang'24) | 5.6x wall-clock overhead without parallel scheduling |
> | Transaction guarantees | SagaLLM (Chang'25) | Only correct reactive planner among 4 frontier LLMs |
> | Multi-file complexity | MAGIS (Tao'24) | Complexity sensitivity: -1.55 (MAGIS) vs -25.15 (GPT-4) |
> | Inter-agent misalignment | Cemri et al. (2025) | 36.9% of 1,642 MAS trace failures from state conflicts |
>
> Two findings are particularly striking. First, CAID shows that adding agents without concurrency isolation actually degrades performance below single-agent levels -- isolation is not optional but necessary for multi-agent benefit. Second, MAGIS's dramatic reduction in multi-file sensitivity (from -25.15 to -1.55) shows that structured concurrent coordination enables handling complex cross-file dependencies that single agents cannot manage. Notably, none of these systems explicitly cite concurrency control literature, yet all reinvent its core patterns -- the quantitative effect of concurrency is already being measured, just not recognized as such.

---

> > ### Author Rebuttal · Reviewer_pxB7 · 2026-04-04
> >
> > I thank the authors for the detailed responses. I have no further questions.

---

> > > ### Author Response · Authors · 2026-04-06
> > >
> > > Thank you again for your thoughtful review and for fully acknowledging our rebuttal responses.
> > >
> > > We are glad that all of your concerns have been adequately addressed. As the discussion period is drawing to a close, we would warmly invite you to consider whether the resolution of your four raised points — the expanded alternative views, the strengthened call-to-action mapping, the explicit related work positioning, and the consolidated quantitative evidence on concurrency effects — might warrant an updated score reflecting the revised contribution.
> > >
> > > We deeply appreciate your time and constructive engagement throughout this process. Thank you very much!

---

### Official Review · Reviewer_NrPh · 2026-03-12

**Significance:** 2
**Argument Clarity:** 3
**Rating:** 5
**Confidence:** 3

**Questions:**

Can the authors be more specific about which kinds of multi-agent settings are mainly limited by concurrency issues, and which ones are still mostly bottlenecked by reasoning, planning, or communication? This could help more to be more concrete for this position idea.

Do the authors have any more preliminary investigation results about some mas failures really come from concurrency problems rather than other causes?

**Alternative Views Section:**

Yes

**Compliance With Llm Reviewing Policy A Conservative:**

Affirmed.

**Discussion Potential:**

2

**Final Justification:**

After reading the authors' rebuttal, my concerns have been addressed clearly. I think the newly added analysis of the related data and prior work is helpful. I also reviewed the newly added tables included in the responses to other reviewers. For researchers who want to work on this topic, a position paper supported by clear prior data and detailed analysis will be valuable.

I encourage the authors to include several of these tables in the main text. Even though this is a position paper, having some well-organized data would still be very helpful.

**Paper Summary:**

This paper makes a clear position statement about multi-agent systems, especially the llm-based ones, should treat concurrency control as a first-class design issue. The main point is that many multi-agent systems' failures that are often described as coordination or communication problems can also be understood as classical concurrency problems, such as stale reads, lost updates, and inconsistent shared state. The paper motivates this view with some examples, connects the mas failure modes to ideas, like the concurrency control.

**Position:**

Yes

**Position In Title:**

Yes

**Related Work:**

3

**Strengths And Weaknesses:**

Strengths:
1. The paper gave a clear position. And multi-llm agents now actually raised more attention, no only the llm researchers, but also the multi-agent system researchers will be very interested to see this topic. When the LLM-based multi-agent systems get more complicated, this is a real issue and definitely worth raising.
2. this paper is also fairly well organized. The core message is stated early and stays consisten, so it is easy to see what the authors are arguing for. I also appreciated that the paper includes the alternative views section and at least tries to engage with other ways of looking at the problem, discuss this problem from different angels.

Weaknesses:
1. My main concern is that the paper is stronger as a perspective piece than as a fully supported argument. The framing is interesting, but most of the support is still illustrative. I did not see enough systematic evidence showing how often important MAS failures are actually concurrency failures, as opposed to broader issues with reasoning, planning, communication quality, or tool use. 2. I also think the scope needs to be tightened a bit. At times the paper seems to suggest that concurrency control is the key missing piece for MAS reliability in general, but that still feels too focused on one aspect. Because in some settings, concurrency is probably important, but in others, it is just one factor among several other concerns.

**Support:**

3

---

> ### Author Rebuttal · Authors · 2026-03-28
>
> We thank the reviewer for the constructive feedback on scope and evidence. We address each concern below. (All new citations will be added in the revised paper.)
>
> **R2-W1: Support is illustrative, not enough systematic evidence.**
> As a position paper, our goal is to identify a theoretical connection and propose a research agenda rather than to conduct experiments. Nonetheless, we agree the current draft understates the available published evidence. We reorganize existing data to show two key findings:
>
> **(a) Independent systems converge on concurrency control without naming it.** Several recent MAS systems have independently adopted classical CC techniques -- though none frame them as such. This convergence is precisely what our position paper explains.
>
> | System | Mechanism (as described) | CC Equivalent | Improvement |
> |:--|:--|:--|:--|
> | CAID (Geng'26) | "Isolated workspaces + merge" | Optimistic CC + commit | +26.7% acc |
> | CodeR (Chen'24) | "Multi-agent task graph" | Dependency scheduling | 22% vs 10% w/o (-55%) |
> | MegaAgent (Wang'24) | "Parallel agent scheduling" | Concurrent scheduling | 800s vs 4505s (5.6x) |
> | SagaLLM (Chang'25) | "Saga pattern + compensation" | Saga transactions | Only correct reactive planner among 4 LLMs |
>
> CAID's ablation is especially telling: without branch isolation (shared workspace), multi-agent scores 55.5% -- actually *worse* than the single-agent baseline (57.2%). Only with git-worktree isolation does it reach 63.3%. This directly shows that concurrency without proper isolation is net-negative. Our contribution is recognizing the shared structure: what these systems solve ad hoc, concurrency control theory addresses systematically.
>
> **(b) Concurrency-attributable failures are a substantial fraction.**
>
> | Failure Mode | Source | Rate | Concurrency Root |
> |:--|:--|:--|:--|
> | Premature submission | Silo-Bench (Zhang'26) | 37.2% | Missing sync barriers |
> | Consensus failure | Silo-Bench (Zhang'26) | 29.9% | Concurrent conflicting states |
> | Inter-agent misalignment | Cemri et al. (2025) | 36.9% | Stale reads, inconsistent state |
> | Coordination overhead | Silo-Bench (Zhang'26) | RCC <=100% | Concurrency scaling penalty |
>
> In Silo-Bench's 301-run analysis, premature submission + consensus failure account for 67.1% of failures. Premature submission is precisely the absence of synchronization barriers; consensus failure is the distributed consensus problem under concurrent independent reasoning. These are coordination protocol failures addressable by concurrency control primitives.
>
> **R2-W2: Scope needs tightening.**
> We agree. Our claim targets MAS where agents concurrently access shared mutable state during long inference windows -- not all MAS universally. Most affected: (1) collaborative coding agents sharing a repository (ChatDev, MetaGPT, MAGIS), (2) game/simulation environments where world state changes during reasoning (VillagerAgent, Voyager), (3) shared-memory/blackboard architectures. Settings with disjoint inputs face minimal concurrency risk. We do not claim concurrency is the sole or dominant failure mode; it is under-recognized and systematically addressable. We will tighten scope language and add a taxonomy distinguishing concurrency-dominated settings from those bottlenecked by reasoning, planning, or communication.
>
> **R2-Q1: Which settings are mainly limited by concurrency?**
> Those where multiple agents modify overlapping state during another agent's reasoning window: shared file systems in coding MAS, shared world state in game environments, shared message buffers in blackboard systems. The key indicator is whether the state can change during the seconds-to-minutes an agent spends reasoning. Even where reasoning is the primary bottleneck, concurrency failures compound errors -- an agent reasoning correctly on stale state still produces incorrect output.
>
> **R2-Q2: Any preliminary investigation results?**
> Multiple ablation studies isolate concurrency's contribution: CodeR loses 55% of resolved issues when its task graph is removed; MegaAgent takes 5.6x longer without parallel scheduling; CAID gains 26.7% from branch isolation+merge. Silo-Bench's coordination cost reaches 100% at N>=50 for high-dependency tasks -- coordination overhead completely negates parallelization. Yu et al. (2026) independently identify memory consistency as a key open challenge from a computer architecture perspective. These convergent findings from ML, SE, DB, and architecture communities provide strong preliminary evidence.

---

> > ### Author Rebuttal · Reviewer_NrPh · 2026-04-01
> >
> > Thanks for the additional evidence, my concerns have been addressed clearly. I think adding these new comparisons and discussions to the paper will make your position clearer and better supported.

---

> > > ### Author Response · Authors · 2026-04-06
> > >
> > > Thank you for your thoughtful feedback and for taking the time to engage with our rebuttal. We sincerely appreciate your positive assessment and your willingness to revise the score.

---

### Official Review · Reviewer_ZXXg · 2026-03-13

**Significance:** 3
**Argument Clarity:** 4
**Rating:** 4
**Confidence:** 2

**Questions:**

1. The authors discuss the trade-off between blocking (pessimistic control) and wasted inference (optimistic control). Have the authors analyzed which approach is more suitable, given typical LLM inference latency and cost?
2. To what extent can LLMs themselves learn concurrency-aware reasoning, for example, via training or prompting, instead of relying on system-level mechanisms?
3. What metrics would best capture concurrency-related inefficiencies, such as wasted inference from transaction aborts?

**Alternative Views Section:**

Yes

**Compliance With Llm Reviewing Policy A Conservative:**

Affirmed.

**Discussion Potential:**

3

**Paper Summary:**

The manuscript seeks to outline a central area in the design of LLM-based multi-agent systems: the role of concurrency control. The authors argue that many failure modes observed in multi-agent systems can instead be interpreted as classical concurrency anomalies such as stale reads, lost updates, and inconsistent shared state. The authors present a general topic rather than a new algorithmic contribution, framing multi-agent failures through the lens of database and distributed systems theory. The paper proposes that MAS frameworks should explicitly incorporate concurrency control mechanisms and discusses a design space spanning system architecture, infrastructure support, and model capabilities.

**Position:**

Yes

**Position In Title:**

Yes

**Related Work:**

4

**Strengths And Weaknesses:**

Strength
1. The paper provides a compelling systems perspective by mapping MAS coordination failures to well-known concurrency anomalies, bridging multi-agent LLM research with database and distributed systems theory, which is valuable and timely.
2. The discussion of isolation levels, optimistic vs. pessimistic control, and MVCC provides a useful taxonomy for MAS design.
3. The paper suggests concrete integration strategies (e.g., Git-based workflows, transactional edits, versioning infrastructure).

Weakness
1. The work is primarily conceptual and does not present experiments or benchmarks demonstrating improved reliability using the proposed concurrency-control perspective.
2. The paper stops short of proposing specific mechanisms or implementations tailored to LLM agents.
3. It remains unclear how to quantitatively evaluate concurrency-aware MAS beyond existing agent benchmarks.

**Support:**

3

---

> ### Author Rebuttal · Authors · 2026-03-28
>
> We thank the reviewer for recognizing the value of bridging MAS and systems/DB theory. We address each point below. (All new citations will be added in the revised paper.)
>
> **R1-W1: No experiments or benchmarks.**
> As a position paper, our contribution is identifying a previously unrecognized theoretical connection and proposing a research agenda -- the ICML Position Track evaluates the strength of argument and potential to shape future research. That said, we will state evidential claims more carefully, and we note a striking pattern in recent work that substantiates our thesis: multiple independent MAS systems have converged on what are, in essence, classical concurrency control techniques -- though none explicitly frame them as such. CAID (Geng & Neubig, 2026) introduces git-worktree isolation + merge-time validation (= optimistic CC), gaining +26.7% accuracy on PaperBench. CodeR (Chen et al., 2024) uses a multi-agent task graph (= dependency scheduling); removing it halves resolution rate (22%->10%). SagaLLM (Chang & Geng, VLDB 2025) applies saga patterns with compensating rollback (= saga transactions), achieving it is the only system to correctly handle reactive planning among four frontier LLMs tested. Our contribution is recognizing this shared structure: what these systems solve ad hoc, concurrency control theory addresses systematically. See also our response to R2-W1 for additional evidence on failure attribution.
>
> **R1-W2: Stops short of proposing specific mechanisms.**
> We propose concrete directions in Sec 3.3: branch-per-subtask isolation, transactional file edits, validation-at-merge, and KV-cache checkpointing for rollback (Sec 3.2). We will strengthen the call-to-action by mapping each to open problems with success criteria: e.g., "Design for latency asymmetry" -> MVCC with KV-cache snapshots; "Build infrastructure" -> transactional APIs for MAS frameworks; "Train for concurrency" -> benchmarks for conflict anticipation/resolution.
>
> **R1-W3: Unclear how to quantitatively evaluate.**
> Existing systems each measure coordination differently (Silo-Bench: relative coordination cost; CAID: pass rate with/without isolation; CodeR: ablation delta). Our framework provides a unified vocabulary: conflict frequency, resolution success rate, wasted computation from aborts, and effective parallelism ratio (Sec 3.4.1). These metrics isolate concurrency-specific overhead from capability failures and can be applied across diverse MAS benchmarks.
>
> **R1-Q1: Pessimistic vs. optimistic control.**
> Neither dominates due to temporal asymmetry (Sec 3.1). CAID empirically validates optimistic control (isolated branches + merge-time validation) for coding tasks. Its ablation is revealing: without branch isolation, multi-agent actually scores *worse* than single-agent (55.5% vs 57.2%) -- only with isolation does it reach 63.3%. Pessimistic locking suits only narrow bottlenecks where the critical section is a tool call (ms), not inference (min). MVCC offers a practical middle ground for moderate contention. We will add a structured comparison in revision.
>
> **R1-Q2: Can LLMs learn concurrency-aware reasoning?**
> An open empirical question (Sec 3.4.2-3.4.3). Pretraining corpora predominantly cover sequential programming; concurrent coordination scenarios (lock denial, abort recovery) are underrepresented. Whether prompting suffices or explicit training is needed has direct design implications: if models cannot reliably learn this, system-level enforcement becomes necessary -- as the convergence pattern in G1(a) demonstrates.
>
> **R1-Q3: Metrics for concurrency inefficiencies.**
> We propose (Sec 3.4.1): (i) conflict frequency under varying contention, (ii) conflict resolution success rate, (iii) wasted computation from aborts (tokens + wall-clock), (iv) effective parallelism ratio. These map to measurements already present in recent work -- e.g., Silo-Bench's RCC metric is precisely our "concurrency overhead" measured as 1-SR(N)/SR(1), reaching 100% at N>=50 for high-dependency tasks.

---

### Decision · Program_Chairs · 2026-04-30

**Decision:**

Accept (regular)

**Comment:**

The paper has real strengths: it is clearly written, raises a timely and potentially important perspective, and the authors engaged seriously with the discussion, with reviewers generally indicating that their concerns were resolved after rebuttal. However, one reviewer maintains substantive concerns about fit and support. The main question is whether the paper makes a sufficiently compelling and well-supported case that merits broad exposure within ICML, and in my judgment the current submission will sufficiently attract interest from the community.